# Oat Extract Avenanthramide-C Reverses Hippocampal Long-Term Potentiation Decline in Tg2576 Mice

**DOI:** 10.3390/molecules26206105

**Published:** 2021-10-10

**Authors:** Yu-Young Lee, Ming Wang, Yurim Son, Eun-Ju Yang, Moon-Seok Kang, Hyun-Joo Kim, Hyung-Seok Kim, Jihoon Jo

**Affiliations:** 1Department of Central Area, National Institute of Crop Science, Rural Development Administration, Suwon 16429, Korea; yurim1023@gmail.com (Y.S.); gr27@korea.kr (M.-S.K.); tlrtod@korea.kr (H.-J.K.); 2Department of Biomedical Sciences, Chonnam National University Medical School, Gwangju 501-757, Korea; Ming322@jnu.ac.kr; 3Research Institute of Pharmaceutical Sciences, College of Pharmacy, Kyungpook National University, 80 Daehak-ro, Daegu 41566, Korea; ejy125@gmail.com; 4Department of Forensic Medicine, Chonnam National University Medical School, Gwangju 501-757, Korea; 5Center for Creative Biomedical Scientists, Chonnam National University Medical School, Gwangju 501-757, Korea

**Keywords:** memory, Alzheimer’s disease, avenanthramide-C, polyphenol, oats, hippocampal synaptic plasticity, amyloid β

## Abstract

Memory deterioration in Alzheimer’s disease (AD) is thought to be underpinned by aberrant amyloid β (Aβ) accumulation, which contributes to synaptic plasticity impairment. Avenanthramide-C (Avn-C), a polyphenol compound found predominantly in oats, has a range of biological properties. Herein, we performed methanolic extraction of the Avns-rich fraction (Fr. 2) from germinated oats using column chromatography, and examined the effects of Avn-C on synaptic correlates of memory in a mouse model of AD. Avn-C was identified in Fr. 2 based on ^1^H-NMR analysis. Electrophysiological recordings were performed to examine the effects of Avn-C on the hippocampal long-term potentiation (LTP) in a Tg2576 mouse model of AD. Avn-C from germinated oats restored impaired LTP in Tg2576 mouse hippocampal slices. Furthermore, Avn-C-facilitated LTP was associated with changes in the protein levels of phospho-glycogen synthase kinase-3β (p-GSK3β-S9) and cleaved caspase 3, which are involved in Aβ-induced synaptic impairment. Our findings suggest that the Avn-C extract from germinated oats may be beneficial for AD-related synaptic plasticity impairment and memory decline.

## 1. Introduction

Alzheimer’s disease (AD) is an irreversible, progressive neurodegenerative disease that is associated with deficits in cognitive function and memory loss [1]. Progressive memory impairment in AD is thought to be encoded by synaptic degeneration [2]. Several hypotheses that have been put forward for AD pathogenesis, the amyloid hypothesis is one of the factors, which give the concept that excessive amyloid beta (Aβ) levels predominantly cause the origin and progression of the nervous system dysfunction and synaptic pathology [3]. The best understood AD pathogenesis in the central nervous system (CNS) could be concluded in a loss of plasticity [4]. As such, measurement of the synaptic plasticity is considered a useful indicator in the evaluation of AD pathophysiology. Synaptic plasticity refers to the long-lasting enhancement of synaptic efficacy induced by trains of electrical stimuli. Several forms of synaptic plasticity have been described in the mammalian central nervous system (CNS). Long-term potentiation (LTP) and other forms of synaptic plasticity are typically regarded as the core cellular substrates underlying learning and memory [5,6]. Electrophysiological recordings of LTP in the mouse hippocampal CA1 region are widely used to investigate synaptic efficacy under normal and pathological conditions [7,8,9]. Developing the therapeutic candidates that against the Aβ toxicity would be beneficial for memory retrieval in AD; one study recently reported that Aducanumab, which is approved by the FDA, could reduce Aβ plaque in AD patients’ brain [10]. Although numerous research groups have targeted the pathogenesis of AD, few of therapeutic agents have been provided. However, natural compounds, due to their antioxidants and anti-inflammatory properties, have been recently documented as promising candidates for the prevention and treatment of AD [11,12,13].

Oats (*Avena sativa* L.) are a healthy and nutritious food containing high concentrations of well-balanced protein, soluble fiber, and energy, as well as a variety of vitamins and minerals [14,15]. The consumption of oats has been reported to provide various benefits such as suppressing host cholesterol, reducing the risk of cardiovascular disease, alleviating diabetes symptoms [16], and preventing obesity [17]. β-glucan is the primary component of oats that is considered to underpin many of these health benefits; however, other constituents of oats may also have beneficial effects. Oats possess a high antioxidant capacity due to the presence of tocopherols, tocotrienols, phytic acid, flavonoids, and phenolic compounds, including avenanthramides (Avns) [18].

Avns comprise three major isoforms, namely, Avn-A, Avn-B, and Avn-C, which belong to the esters of 5-hydoroxyanthranilic acid with *p*-coumaric (Bp), ferulic (Bf), and caffeic acid (Bc), respectively [19]. The antioxidant activity of Avns is 10–30 times greater than that of other typical cereal components, such as gentisic acid, phydroxybenzoic acid, protocatechuic acid, syringic acid, ferulic acid, and vanillin [20,21,22,23]. Additionally, the beneficial anti-inflammatory, anti-hypertensive, and anti-atherogenic effects of Avns are well-documented [24,25]. Among the Avns, the Avn-C constituent in oats is two-fold higher than that of Avn-A and Avn-B [26]. The antioxidant capacity of Avn-C has been reported to closely resemble that of the standard synthetic antioxidant, butylated hydroxytoluene (BHT), in β-carotene. Moreover, Avn-C has a higher bioactivity compared with 6-hydroxy-2,5,7,8-tetramethylchromane-2-carboxylic acid (Trolox) in DPPH [27].

Recent evidence suggests that synthetic Avn-C exerts a protective role on synaptic impairment in AD pathology [28]. Given the potential applicability of harnessing this treatment for AD, in this study, we performed methanolic isolation of Avn-C from sprouted oats to evaluate the effects of purified Avn-C on synaptic plasticity in a mouse model of AD. Our findings suggest that Avn-C reverses hippocampal LTP decline, highlighting its therapeutic potential as a treatment for memory impairment in AD.

## 2. Results and Discussion

### 2.1. Avn Content in Different Oat Cultivars

The Avn content in nine Korean cultivars of oats is presented in Table 1. The total Avn content ranged from 2.9–188.7. Oat samples from DY contained the highest levels of all types of Avns (Avn-C, Avn-A, and Avn-B), whereas SH seeds contained the lowest levels of total Avns. Among the oat cultivars, Avn content was higher in DY, JM, SY, SE, and CY from naked oats. However, the total Avn content was lower in DH, JP, HS, and SH from hulled oats, in agreement with previous reports [29]. Avn-C content varied notably among the different oat cultivars, with the highest value observed in DY (86.9 ± 24.1 μg/g, *p* < 0.01). In agreement with our findings, a previous study reported that the mean Avn content was 65.7 mg/kg and 62.2 mg/kg in Belle and Gem, respectively, which was higher than that in Dane (37.1 mg/kg), and that the Avn quantity in oats was influenced by genotype and environmental factors [30]. Indeed, Switzerland produces oats with a lower Avn content compared with Canada [20].

Germinated oat samples from DY were selected for Avn extraction. The mean Avn content decreased slightly after germination for 24 h, but increased significantly after germination for 48 h and 72 h, respectively (*p* < 0.01) (Table 2). The Avn-C and total Avn contents in germinated oats were approximately 2.5-fold higher than those in raw oats. The total phenol content and antioxidant activity in the procedure-rich fraction are presented in Table 3. These findings are consistent with those of a previous report that the Avn content of germinated oats is higher in Vista, Gem, and Dane oat cultivars than in raw oats [31].

### 2.2. Identification of Avn-C Content in Fr. 2

Methanolic extraction of the Avn-rich fraction (Fr. 2) from germinated oats was performed using column chromatography. Avn-C was identified in Fr. 2 based on ^1^H-NMR analysis. Figure 1 illustrates the Avn-C extraction procedure. In total, 31 mg/g of total Avns and 0.52 ± 0.02% of β-glucan were obtained from Fr. 2. Fr. 2 contained high levels of Avn-C (99.9% of total Avns), but only low levels of Avn-A and Avn-B were detected (Table 4). The Avn-C content of Fr. 2 was 16-fold higher than that of raw oats. The β-glucan content of Fr. 2-2 was approximately five-fold lower than that of Fr. 2.

### 2.3. Isolation and Purification of Avn-C

The purity of isolated compound 1 from Fr. 2 was evaluated by spectrum (Figure 2 and Figure 3) and was confirmed as Avn-C by ^1^H-NMR analysis (Figure 4).

Compound **1**: light green powder; ^1^H-NMR (500 MHz, MeOH-*d*_4_): δ8.45 (1H, d, *J* = 9.0 Hz, H-3), 7.52 (1H, d, *J* = 3.2 Hz, H-6), 7.50 (1H, d, *J* = 15.6 Hz, H-7’), 7.07 (1H, d, *J* = 1.7 Hz, H-2’), 7.02 (1H, dd, *J* = 9.0 and 3.2 Hz, H-4), 6.97 (1H, dd, *J* = 8.2 and 1.7 Hz, H-6’), 6.79 (1H, d, *J* = 8.2 Hz, H-5’), and 6.46 (1H, d, *J* = 15.6 Hz, H-8’) (Figure 3). From these data, compound 1 was postulated as Avn-C and this was finally confirmed by a comparison of these results from the ^1^H-NMR analysis with those of a previous study [32].

### 2.4. Restoration of HFS-Induced LTP by Avn-C

Previous studies have reported synaptic plasticity deficits in a well-characterized Tg2576 amyloid precursor protein (APP) transgenic mouse model, including Aβ deposition and impaired LTP in the hippocampal CA1 region [33]. In order to examine the potential function of Avn-C on Aβ-caused synaptic impairment, we used hippocampal LTP analysis, the most studied form of synaptic plasticity. Before recording the LTP, tg2576 mouse hippocampal slices were incubated with extracted Avn-C (50 μM) for 2 h. Then, the LTP recordings were conducted at CA3 to CA1 synapses, and LTP was induced by high-frequency stimulation (HFS; two 100-Hz stimulations at a 30-s interval), and field excitatory postsynaptic potentials (fEPSPs) were recorded for at least 1 h. Assessment of LTP showed that HFS-induced LTP was impaired in non-treated Tg2576 mouse hippocampal slices (120.4 ± 2% of baseline, *n* = 5). Remarkably, Avn-C application restored LTP impairment in the hippocampus of Tg2576 mice (145.9 ± 9% of baseline, *n* = 6), accompanied by an increased fEPSP compared with that of the control (Figure 5a). These data indicated that Avn-C may exert a protective role on synaptic plasticity deficit in Tg2576 mice brain, leading to the improvement of neuronal function against Aβ pathogenesis.

AD brains contain excessive Aβ levels linking the hallmarks to synapse damage and disease progression [34]. Even though the accurate mechanisms of Aβ impairs synaptic function are not fully understood, it is believed that acute exposure of hippocampal slices to Aβ inhibits LTP by activating caspase-3 and GSK3β [35]. To gain further insight into the mechanisms of enhancement of LTP in Avn-C-treated Tg2576 mice, we performed Western blotting experiments on hippocampal slices in the presence of Avn-C. Protein levels of p-GSK3β (S9) were increased and those of cleaved caspase-3 (Clv. C3) were decreased in the Avn-C treated hippocampus (Figure 5b), suggesting that Avn-C works as an inhibitor to suppress the abnormal activities of GSK3β and caspase-3 in an AD model mouse brain. This further supports that the mechanism underlies Avn-C rescues Aβ-inhibited LTP by modulating GSK3β and caspase-3 expressions.

It is believed that abnormal activities of caspases and GSK-3 are also highly related to the neuropathology of AD [36,37]. In transgenic AD models, Aβ activates GSK-3β via inhibiting the phosphorylation of this enzyme [38]. The activation of this enzyme has been reported in the postmortem of AD patients [39]. This evidence indicates that the fine-tuning of caspase-3 and GSK-3β inhibit Aβ-induced pathology and LTP impairment under the AD condition. In our finding, Avn-C exerts a protective effect on Aβ-induced synapse failure by inhibiting GSK-3β. Consistently, GSK-3β inhibition has been shown to reduce Aβ production in AD models [40].

## 3. Materials and Methods

### 3.1. General Apparatus and Chemicals

Column chromatography was performed using Kiselgel 60 (Merck, Art. 7734, Art. 9385, NJ, USA) and RP-18 (YMC, Art. 14878, Japan). All purifications were monitored using thin-layer chromatography (TLC) precoated with Kiselgel 60 F_254_ (Merck, Art. 5715). Compound purifications were performed using a high-performance liquid chromatography (HPLC) core system (Thermo Scientific, Germering, Germany; Dionex Ultimate 3000 Diode Array with ELSD at 254, 280, and 365 nm; and a Dionex Ultimate 3000 pump). A Kinetex 5 μm C18 100 Å (250 × 4.6 mm, Phenomenex, CA, USA) column was used for isolation. The mobile phase consisted of solvent A (water with 0.1% acetic acid) and solvent B (MeOH with 0.1% acetic acid) at a flow rate of 1.0 mL/min. The injection volume was 10 μL, and the column temperature was 35 °C. The ^1^H-NMR spectrum was recorded using a Bruker Avance Digital 500 spectrometer (Karlsruhe, Germany) at 500 MHz. Chemical shifts were provided as δ (ppm) from tetramethylsilane (TMS). All of the reagents were purchased from Sigma-Aldrich (St. Louis, MO, USA).

### 3.2. Oat Samples

Five naked and four covered oat cultivars were used for the analysis of Avns (Table 5). All oats were planted in October and harvested in July between 2014 and 2016 at the National Institute of Crop Science, Rural Development Administration (RDA) Farm, Iksan, Korea. The oats were stored at 14 °C for 1 month. The naked oats were subjected to a selection process, given the ease of separating the hull from the grain. The outer hull of the covered oats was separated using a short stalk (FC2K, Yamamoto, Japan).

For germination, the cultivar “Daeyang,” which has the high Avn content among Korean oat cultivars, was selected. Oat seeds (500 g) were germinated in plastic dishes with fine holes using the modified soybean germinating utility (temperature of 21 °C, treated with artificial rainfall for 5 min, and stopped for 20 min) for 72 h. The oats were collected at 24, 48, and 72 h and then dried at 40 °C for 24 h. All of the germinated oat samples were stored at −70 °C until laboratory extraction and analysis. All of the samples were ground at 260 g for 200 s using an auto-mill disintegrator (Tokken, Japan) in a liquid nitrogen environment.

### 3.3. Sampling of Avns-Rich Fractions

#### 3.3.1. Extraction

Germinated oat flour (1000 g) was defatted with 4 L of n-hexane (500 mL × 2) for 7 days at room temperature. The precipitates were placed in a fume hood for 24 h to completely remove the hexane. Then, 1000 g of defatted flour was added to 1 L of 80% EtOH at 28 °C for 16 h in a shaking incubator (240 rpm). Ethanol extract (80%, 500 g) was filtered using Buechner funnels (Duran, Mitterteich, Germany), and the precipitates were re-extracted with 80% ethanol (1 L) at 50 °C for 30 min. The process was repeated twice and then evaporated at 40 °C. The extract (39.0 g) was suspended in methanol (200 mL) and was sonicated for 10 min. After centrifugation for 5 min at 6225 g, the supernatant was evaporated and collected.

#### 3.3.2. Purification

Avns-rich fractions from oats were chromatographed on silica gel and RP-18 using a solvent system. The fractions were confirmed to have the same Rf value of Avns using normal-phase TLC. The methanol extract (30.5 g) was chromatographed on silica gel (230–400 mesh, Ø 7.5 × 35 cm) using a chloroform/methanol/water solvent system under isocratic conditions (10:7:1) to yield 60 fractions (Fr. 1–60) (Figure 1). Each fraction was subjected to normal-phase TLC under the same solvent conditions. Fr. 2 (Fr. 27–36) was confirmed to have the same Rf value as that of Avn-C. The content of Avn-C was analyzed using HPLC.

The subfraction 2-2 (1.03 g) was chromatographed using ODS column (150 μm, Ø 2.8 × 35 cm, stepwise gradient of 40–100% MeOH) chromatography to yield 30 subfractions (SFr. 1–SFr. 30). The subfraction 2-2 (1.03 g) was subjected to HPLC (Luna 5μ, C18, 100 Å 10 × 250 mm, Phenomenex, Torrance, CA, 0.1% HOAc in 40% MeOH, 3.0 mL, min^−1^) system for purification and 2.1 mg of purified compound 1 was obtained from subfraction 16-20.

#### 3.3.3. Analysis of Avns and β-Glucan Content

Finely ground oats (1 g) were extracted with 100 mL of 80% EtOH in 10 mM phosphate buffer (pH 2.8) in a shaking incubator (240 rpm) for 16 h at 37 °C; this process was repeated three times. The extracts were collected as a filtrate (No. 2) and were evaporated at 50 °C in a Syncore Analyst (Buchi, Switzerland). Dried samples were resuspended in 2.0 mL of 80% EtOH and filtered through a 0.2-μm filter unit (Millipore, Bedford, MA, USA) prior to ultra-performance liquid chromatography (UPLC) analysis.

The UPLC system was equipped with a photodiode array detector (Waters, Milford, MA, USA) with detection at 340 nm and an Acquity UPLC^®^ HSS C18 column (2.1 × 100 mm, 1.8 μm; Waters). The mobile phase comprised buffer A (0.01 M phosphate buffer, pH 2.8) and buffer B (100% acetonitrile). A gradient of 15–40% of buffer B was applied over 9 min at a flow rate of 0.6 mL/min. The injection volume was 1 μL, and the column temperature was adjusted to 35 °C. For quantification and identification purposes, standard stock solutions of Avn-A, Avn-B, and Avn-C were prepared in dimethyl sulfoxide. Avn peaks were identified by comparing the sample retention times with the standards. The concentrations were calculated from the peak areas determined through linear regression. All extracts were obtained from three independent samples.

The β-glucan content of the oat seeds and extracts was analyzed using an enzyme test kit (Megazyme International Ireland Ltd., Bray Business Park, Bray, Ireland). Briefly, 0.1 g of sample was mixed with 200 μL of 50% EtOH and 4 mL of 20 mM sodium phosphate buffer (pH 6.5) by vortexing. The sample was immediately incubated at 100 °C for 1 min; this process was repeated twice. The extract was incubated at 50 °C for 5 min and then mixed with 200 μL of lichenase. After 1 h of reaction and cooling, 5 mL of 200 mM sodium acetate buffer (pH 4.0) was added to the mixture, followed by centrifugation at 3000 rpm for 10 min. A volume of 100 μL of the supernatant was added to 100 μL of β-glucosidase and incubated at 50 °C.

#### 3.3.4. Total Phenol Quantification, DPPH, and ABTS

Total polyphenol quantification was performed with spectrophotometric analysis using the Folin–Ciocalteu phenol reagent. The extract (100 μL) was mixed with 2% Na_2_CO_3_ solution (2 mL) and incubated for 3 min at room temperature. Then, 100 μL of 50% Folin–Ciocalteu phenol reagent was added. After 3 min of incubation, the reactants were transferred to a 96-well plate, and the polyphenol content was determined at 750 nm. Gallic acid was used as a standard, and the results were expressed as mg gallic acid equivalent (GAE)/g of extract.

Extracts (200 μL) in the cuvette were added to 800 μL of DPPH solution (0.2 mM in ethanol) and incubated for 30 min at 25 °C to avoid exposure to light. The antioxidant effects were determined by measuring the absorbance at 515 nm using a spectrophotometer (Varioskan LUX, Thermo Scientific). To prepare the ABTS stock solution, 7 mM ABTS was mixed with 2.45 mM potassium persulfate and incubated in the dark for 24 h. The stock solution was diluted with methanol to obtain an absorbance between 1.4–1.5 at 735 nm. Diluted ABTS (1 mL) was added to 50 μL of the sample and incubated for 10 min. The absorbance was measured at 735 nm. DPPH radical and ABTS cation radical scavenging activities were expressed in terms of Trolox equivalent (TE) per gram of extract.

### 3.4. Animals

Tg2576 male mice (APP KM670/671NL, Taconic; 9 to 10 months of age) were used for the experiments. The animals were individually housed in ventilated cages with ad libitum access to food and water. The breeding room was controlled with a 12-h light/dark cycle, and the temperature was maintained at 22–30 °C. The animals were sacrificed between 09:00 a.m. and 10:00 a.m. by cervical dislocation. All of the experiments involving animals were performed in accordance with protocols approved by the Institutional Animal Care and Use Committee of Chonnam National University Medical School.

### 3.5. Hippocampal Slice Preparation

After cervical dislocation, the brains were rapidly removed, transferred to ice-cold artificial cerebrospinal fluid (aCSF; 124 mM NaCl, 3 mM KCl, 26 mM NaHCO_3_, 1.25 mM NaH_2_PO_4_, 2 mM CaCl_2_, 1 mM MgSO_4_, and 10 mM glucose), and constantly perfused with a gas mixture of 95% O_2_ and 5% CO_2_. A midsagittal cut was made in the brains and one hemisphere was kept in ice-cold aCSF until it was required. Hippocampal slices were cut transversely (400-μm thick) using a McIlwain tissue chopper (Mickle Laboratory Engineering Co. Ltd., Surrey, UK) and were stabilized for 1 h in aCSF at room temperature.

### 3.6. Electrophysiology

Hippocampal slices were treated with 50 μM Avn-C for 2 h prior to recordings. Stimulating electrodes were placed in the Schaffer collateral pathway. Extracellular field potentials were recorded in the CA1 region using microcapillary electrodes containing 3 M NaCl. Stimuli were delivered alternately to the two electrodes (0.016 Hz for each electrode). After establishing a stable baseline for 30 min, LTP was induced by high-frequency stimulation (HFS; two 100-Hz stimulations for 1 s at a 30-s interval), and field excitatory postsynaptic potentials (fEPSPs) were recorded for at least 1 h. The slope of the evoked field potential response was measured and expressed relative to the normalized baseline. Data were collected using an NI USB-6251 data acquisition module (National Instruments, Austin, Texas, USA), amplified by an Axopatch 700 B amplifier (Axon Instruments, Foster City, CA, USA), and captured and analyzed using WinLTP 2.32 software (www.winltp.com, accseed on 6 September 2021).

### 3.7. Immunoblotting

Hippocampal tissue lysates were prepared using a radioimmunoprecipitation assay buffer (Cell Biolabs, Inc., San Diego, CA, USA). Protein concentrations were determined using a bicinchoninic acid (BCA) assay. The proteins were separated on 10–12% gels and transferred to polyvinylidene fluoride (PVDF) membranes (Millipore, Bedford, MA, USA), which were incubated overnight with primary antibodies. The following primary antibodies were purchased from Cell Signaling Technology (MA, USA): anti-cleaved caspase-3 (1:1000), anti-phospho-GSK-3β (Ser9) (p-GSK3β-S9) (1:1000), and anti-β-actin (1:1000). Membranes were then incubated with rabbit IgG antibodies (1:3000, Cell Signaling Technology) conjugated to horseradish peroxidase and immunoblotted using an enhanced chemiluminescence (ECL) detection system (Millipore, Bedford, MA, USA). The optical density of the immunoreactive bands was measured using ImageJ software (National Institutes of Health, Bethesda, MD, USA).

### 3.8. Statistical Analysis

All experiments were performed in triplicate. The data are expressed as the mean ± standard error of the mean (S.E.M.). Statistical significance was determined using one-way ANOVA. Statistical analyses were performed using GraphPad Prism 7 (GraphPad, La Jolla, CA, USA) and SAS version 9.2 (SAS Institute, Cary, NC, USA). Means were compared using Duncan’s test or an unpaired Student’s *t*-test. Statistical significance was set at *p* < 0.05.

## 4. Conclusions

Alzheimer’s disease is a neurodegenerative disorder and is the most prevalent form of dementia all over the world. AD attacks the central nervous system and leads to synapses loss, cognitive deficit, and memory decline. It is associated with pathological accumulation of Aβ plaques in the brain. The increasing number of AD incidence require new therapeutic agents [41]. Among them, natural compounds have recently been paid attention and their potential effects on AD pathology have been well documented, for example, vitamin C reduces Aβ plaques in an AD mouse model [42], Bryostatin extracted from *bryozoan Bugula neritina* inhibits the Aβ accumulation and enhances memory function in AD mouse model [43], and Resveratrol is a natural compound present in grapes that exerts clearance effect on Aβ aggregates [44]. This study demonstrated for the first time that Avn-C, extracted from germinated oats, prevented the impairment of synaptic plasticity in Tg2576 mice and abnormal activation of GSK3β under AD pathologies. Our findings suggest Avn-C as a potential therapeutic candidate for AD-related synaptic plasticity impairment and memory decline. Nevertheless, we cannot disregard other factors that also involve in AD progression, such as the known hyperphosphorylated tau protein and the neuroinflammation in the central nervous system [45]. Accumulation of p-tau protein in the neurofibrillary tangles accompanied by an increase of GSK3β activity [46] directed us to consider the potential role of Avn-C on tau pathology. Additionally, the anti-inflammatory benefits [47] of Avn-C have been reported [25], therefore, the anti-neuroinflammation effect of Avn-C in AD remains to be further evaluated.

## Figures and Tables

**Figure 1 molecules-26-06105-f001:**
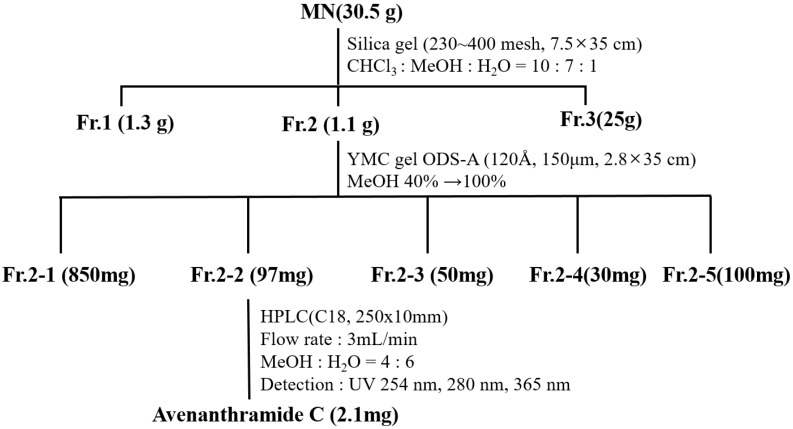
Procedure of Avn-C extraction. MN, methanolic extract; HPLC, high-performance liquid chromatography. Fr. 2, Avn-C-rich fraction; Avn-C, avenanthramide C; Avn-A, avenanthramide A; Avn-B, avenanthramide B.

**Figure 2 molecules-26-06105-f002:**
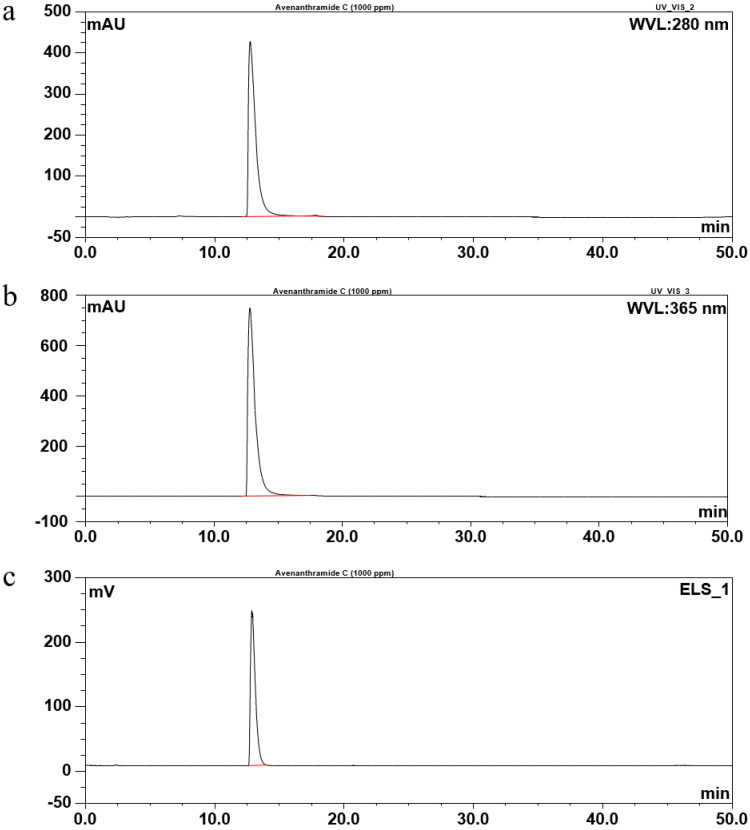
HPLC chromatograms of Avn-C isolated from Fr. 2. Avn-C was detected by UV 254 nm (**a**), UV 365 nm (**b**), and ELSD (**c**).

**Figure 3 molecules-26-06105-f003:**
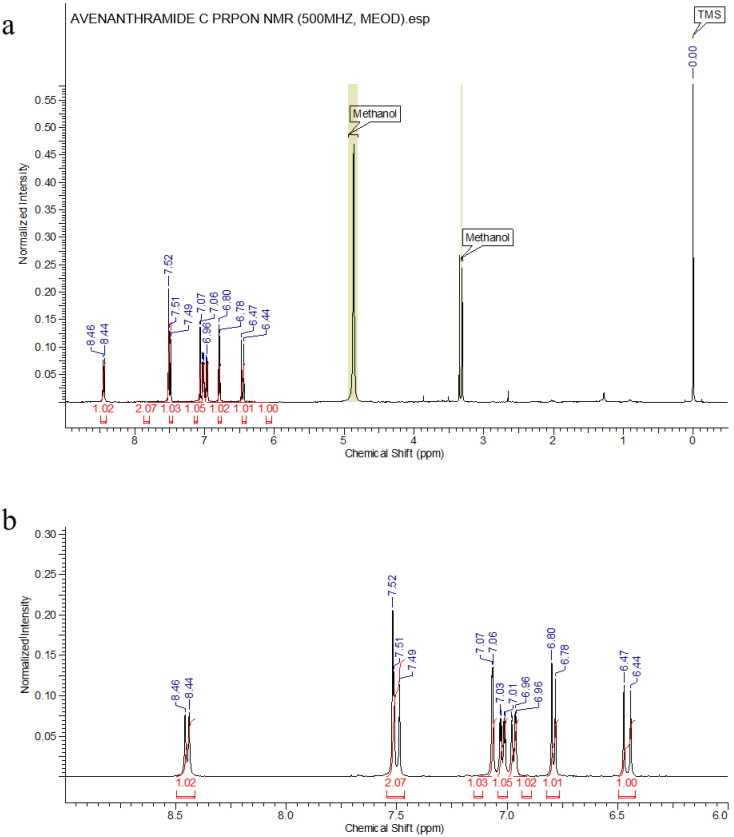
^1^H NMR spectra of Avn-C (MeOH-d4). (**a**) The whole spectrum. (**b**) The magnified spectrum from 9.0 to 6.0 ppm.

**Figure 4 molecules-26-06105-f004:**
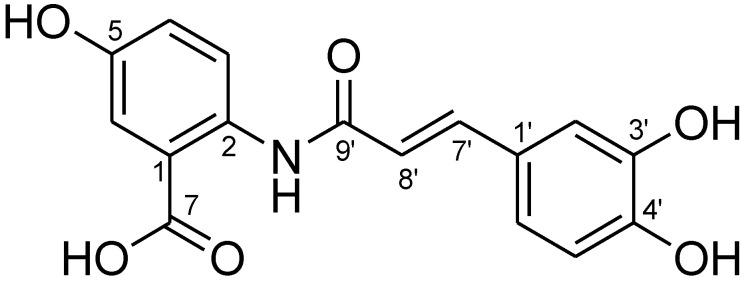
The chemical structure of compound 1 (Avn-C) isolated from germinated oats.

**Figure 5 molecules-26-06105-f005:**
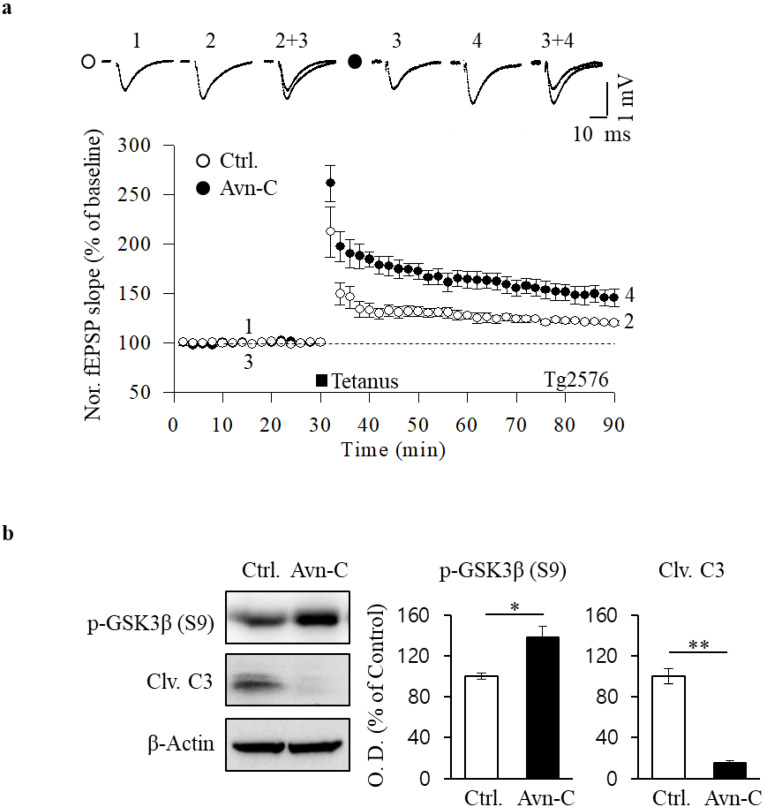
Avn-C restores LTP in Tg2576 mouse hippocampus. (**a**) LTP induction was triggered by high-frequency stimulation (HFS: 2× tetanic stimulation at 100 Hz, 100 pulses; interstimulus interval, 30 s) in the hippocampal CA1 region of Tg2576 mice. The control group (open circles, 120.4 ± 2% of baseline) and Avn-C incubated group (closed circles, 145.9% ± 9% of baseline); *n* = 5 to 6 per group from 5 to 6 animals. (**b**) Representative Western blots (left) and group data (right) indicating altered p-GSK3β (S9) and cleaved caspase-3 expression (Clv. C3) in the 50 μM Avn-C-treated group compared with that in the control group; *n* = 4 per group from 4 animals. Data are expressed as mean ± S.E.M. Differences were considered significant at * *p* < 0.05, ** *p* < 0.01. Ctrl., control; fEPSP, field excitatory postsynaptic potentials; O. D., optical density; Avn-C, Avenanthramide C; LTP, long-term potentiation.

**Table 1 molecules-26-06105-t001:** Comparison of Avn contents in oat cultivars (μg/g).

Type	Cultivars	Abbr.	Avn-C	Avn-A	Avn-B	Total Avn
Naked	Seonyang	SE	11.7 ± 2.5 ^b^	13.7 ± 3.5 ^b^	13.4 ± 3.9 ^b^	38.9 ± 8.8 ^b^
	Daeyang	DY	86.9 ± 24.1 ^a^	52.7 ± 16.7 ^a^	49.1 ± 24.3 ^a^	188.7 ± 85.1 ^a^
	Choyang	CY	7.8 ± 1.7 ^b^	8.6 ± 2.4 ^b^	10.5 ± 6.1 ^b^	27.0 ± 10 ^b^
	Suyang	SY	15.9 ± 0.7 ^b^	14.3 ± 2.7 ^b^	21.1 ± 9.5 ^b^	51.3 ± 11.6 ^b^
	Jungmo2005	JM	22.9 ± 6.8 ^b^	26.4 ± 8.3 ^ab^	18 ± 3.1 ^b^	67.4 ± 18.2 ^b^
Hulled	Samhan	SH	1.6 ± 0.2 ^c^	0.0 ± 0.1 ^c^	1.3 ± 0.1 ^c^	2.9 ± 0.4 ^c^
	Donghan	DH	3.4 ± 1.1 ^c^	3.7 ± 1.0 ^c^	7.5 ± 1.9 ^bc^	14.7 ± 3.9 ^c^
	Highspeed	HS	3.3 ± 0.1 ^c^	2.4 ± 0.1 ^c^	3.6 ± 0.1 ^c^	9.3 ± 0.3 ^c^
	Jopung	JP	3.6 ± 0.1 ^c^	3.0 ± 0.0 ^c^	3.5 ± 0.1 ^c^	10.1 ± 0.2 ^c^
				**	*	**

Data represent the means of three replicates ± S.E.M. ^a,b,c^ Means values in the same raw with different superscripts are significantly different at * *p* < 0.05, ** *p* < 0.01 according to Ducan’s multiple range test. Avn-C, avenanthramide C; Avn-A, avenanthramide A; Avn-B, avenanthramide B; Abbr., abbreviation.

**Table 2 molecules-26-06105-t002:** Changes in Avn content in germinated oats (μg/g).

Germination Time (h)	Avn-C	Avn-A	Avn-B	Total Avns
0	84.3 ± 3.1 ^b^	64.1 ± 2.3 ^a^	58.1 ± 2.2 ^a^	206.5 ± 7.6 ^a^
24	75.5 ± 1.7 ^c^	49.9 ± 0.6 ^a^	46.4 ± 1.2 ^a^	171.9 ± 3.4 ^a^
48	216.0 ± 1.9 ^a^	165.2 ± 2.2 ^a^	115.5 ± 1.4 ^a^	496.7 ± 4.0 ^a^
72	218.8 ± 4.0 ^a^	173.2 ± 2.9 ^a^	129.5 ± 4.2 ^a^	521.5 ± 11.0 ^a^

Data represent the means of three replicates ± S.E.M. ^a,b,c^ Means values in the same raw with different superscripts are significantly different at *p* < 0.05 according to Ducan’s multiple range test. Avn-C, avenanthramide C; Avn-A, avenanthramide A; Avn-B, avenanthramide B Avn, av-enanthramide.

**Table 3 molecules-26-06105-t003:** Changes in the antioxidant activity of rich fraction extracts.

Extract (1 g)	DPPH (mg TE/100 g of Sample)	ABTS (mg TE/100 g of Sample)	Polyphenol (mg GAE/g of Extract)
Germinated grain	11.02 ± 1.23 ^b^	30.76 ± 2.17 ^b^	30.49 ± 2.39 ^b^
MN	9.05 ± 0.32 ^c^	25.35 ± 0.29 ^c^	27.40 ± 0.60 ^c^
Fr. 2	29.77 ± 0.94 ^a^	37.11 ± 0.21 ^a^	44.01 ± 1.32 ^a^

The effects of fraction extract on antioxidant activity were determined using DPPH, ABTS, and Folin-Ciocalteau assays. The data are presented as mg of TE per 100 g (DPPH and ABTS) of extract and mg of GAE per g of the extract. The data are presented as mean ± S.E.M. Values with different superscripts are significantly different at *p* < 0.05 based on one-way ANOVA and Ducan’s multiple range test. TE, Trolox equivalent; GAE, Gallic acid equivalent; MN, methanolic extract; Fr. 2, Avn-C-rich fraction.

**Table 4 molecules-26-06105-t004:** Avn-C contents in Fr. 2 (mg/g).

Extract	Weight (g)	Avn-C	Avn-A	Avn-B	Total Avns	Yield (%)	β-glucan % (g/100 g)
Fr. 2	1.1	31.11	0.01	0.00	31.12	0.2	0.52 ± 0.02

Fr. 2, Avn-C-rich fraction; Avn-C, avenanthramide C; Avn-A, avenanthramide A; Avn-B, avenanthramide B.

**Table 5 molecules-26-06105-t005:** Characteristics of the oat cultivars used in this study.

Cultivar	Abbr.	Registration Year	Type	Utilization Type	Grain Yield (MT ha^−1^)
Seonyang	SE	2003	Naked	Food	3.38
Daeyang	DY	2007	Naked	Food	4.18
Choyang	CY	2007	Naked	Food	4.67
Suyang	SY	2010	Naked	Food	4.35
Jungmo2005	JM	2010	Naked	Food	4.35
Samhan	SH	2001	Hulled	Forage use	4.44
Donghan	DH	2001	Hulled	Forage use	4.21
Highspeed	HS	2005	Hulled	Forage use	5.00
Jopung	JP	2009	Hulled	Forage use	5.68

SE, DY, CY, SY, and JM of grain yield were tested at four locations: Gimjae, Iksan, Jeongeup, and Jinju for 3 years. SH and DH grains were tested in Suwon, Unbong, and Iksan. JP was tested in Gimjae in 2014. The HS of grain yield was tested only at the Jeju location. Abbr., abbreviation; MT., Metric Ton.

## Data Availability

The data presented in this study are available upon request from the corresponding author.

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
