# Peer review of "Oat Extract Avenanthramide-C Reverses Hippocampal Long-Term Potentiation Decline in Tg2576 Mice"

_molecules, 2021, doi:10.3390/molecules26206105_

Round 1
Reviewer 1 Report
I suggest some changes to improve the manuscript.
- The discussion section is missing. I suggest discussing the results and how they can be interpreted from previous studies and the working hypotheses. I recommend mentioning the limitations of the work and future research directions.
- In the introduction and discussion section, I suggest adding in a detailed way the possible mechanism of action involved in the protective role on synaptic impairment in AD pathology.
- It is not clear why the oats must be germinated and why fraction 2 is also fractioned.
- How many animals were used for control and Avn-C treatment?
- In figure 1, fraction 2.2 is repeated. Is that right?
- I suggest improving figure 2 quality.
- I suggest explaining in detail the figures and results of section 2.4 to understand the contribution of avenanthramide-C.
Author Response
- Reviewer #1:
1) The discussion section is missing. I suggest discussing the results and how they can be interpreted from previous studies and the working hypotheses. I recommend mentioning the limitations of the work and future research directions.
Response: We would like to appreciate the reviewer for careful and thorough reading of our manuscript. As suggested by reviewer, we have added the suggested content to paragraph ‘Results and Discussion’ in the manuscript.
As suggested by the reviewer, we have also added the limitations and future research directions of the work in the paragraph ‘Conclusion’. Accordingly, the revised manuscript has been systematically improved with new information and additional interpretations.
2) In the introduction and discussion section, I suggest adding in a detailed way the possible mechanism of action involved in the protective role on synaptic impairment in AD pathology.
Response: We appreciate the reviewer’s suggestion. According to the reviewer’s comments, we have provided more details in the revised manuscript (Introduction: page no. 1, line 38-43) (Results and Discussion: page no. 8, line 186-188, line 197-204).
3) It is not clear why the oats must be germinated and why fraction 2 is also fractioned.
Response: Oat germination has been reported to modify the composition of macronutrients and induce the activity of micronutrients, such as polyphenols. Additionally, in our previous study, Avn-C content was increased from germinated oat compare to that of intact seed (Lee J.H. et al. J.Food Biochem. 2019;43:e12799. https://doi.org/10.1111/jfbc.12799, Kim M et al. J Korean Soc. Food Sci. Nutr. 2019; 48(12) 1337-1344).
why fraction 2 is also fractioned: As described in Figure 1 and 3.3.2. Purification, the column chromatographies were performed for isolation and purification of target compound, Avn-C. The Avns-rich fraction (MN, 30.5 g) means the fraction containing high-contents of avenanthramide “compounds”, not the single compound Avn-C. That’s why the MN was subjected to the column chromatographies. Each subfraction derived from Fr. 2 had different compound constituents with different polarities. Among them, the fraction having the highest content of Avn-C was Fr. 2-2. Therefore, Fr. 2-2 was applied for the last column chromatography to obtain the high purity of Avn-C.
4) How many animals were used for control and Avn-C treatment?
Response: Nine of non-treated Tg2576 mice and Ten of Avn-C treated Tg2576 mice were used in our electrophysiological recording and western blotting studies.
5) In figure 1, fraction 2.2 is repeated. Is that right? I suggest improving figure 2 quality.
Response: We apologize for the mistake in the figure 1 and we have changed it in the revised manuscript.
We have improved the figure 2 quality in the revised manuscript.
6) I suggest explaining in detail the figures and results of section 2.4 to understand the contribution of avenanthramide-C.
Response: Thank you for the suggestion. We have added more information in the figures and results of section 2.4. (page no. 7, line 161-168, line 170-173) (page no. 8, line 191-194).
Reviewer 2 Report
Oat extract avenanthramide-C reverses hippocampal long-term potentiation decline in Tg2576 mice
This manuscript focuses on the methanolic extraction of the Avnsrich fraction (Fr. 2) from sprouted oat using column chromatography and examined the effects of Avn-C on synaptic correlates of memory in a mouse model of AD. Avn-C was identified in (Fr. 2) based on 1H-NMR analysis.
The antioxidant activity of Avns is 10–30 times greater than that of other typical cereal components such as gentisic acid, phydroxybenzoic acid, protocatechuic acid, syringic acid, ferulic acid, and vanillin.The anti-inflammatory, anti-hypertensive, and anti-atherogenic effects of Avns are well documented.
This study demonstrated for the first time that Avn-C, extracted from germinated oat, prevented the impairment of synaptic plasticity in Tg2576 mice and abnormal activation of GSK3β under AD pathologies. This findings suggest that Avn-C is a potential therapeutic candidate for AD-related synaptic plasticity impairment and memory decline
The study is a meaningful supplement to the series of publications regarding the various substance classes and their "synthesis, structure and potential antitumor activity of some corresponding gold(I) and silver(I) complexes or heterocyclic phos-phorous compounds ".
The Chapter Results and Discussion is of good scientific quality and the rich and instructive graphic realizes the understanding of the obtained results and of their significance.
The experimental data is described appropriately and the manuscript needs no language and grammar corrections. The manuscript is written straight forward.
The structure of the compounds were determined using NMR spectroscopy. The characterisations (H-NMR) are described. This is acceptable for organic compounds.
The autors did not reflect any other field of other anti-cancer research, and another derivatives with the same anticancer activity, based on their antioxidant activity.
Still I believe that you should describe in the introduction more generally the synthesis and the use of phosphorous-organic compounds in general as alkylating or antioxidant agents. Therefore, you should cite more special topics. It is of interest for synthetic chemists which wide use have phosphorous-organic compounds. In the work of Schmutzler et al. could be found a huge variation of phosphorus organic compounds and their biological potential.
Because the authors have been presented in the References part of the manuscript a series of scientific papers, that describe a lot of other derivatives, respectively their cytostatic and antimicrobial activity, the authors well have also to present in the introduction part the data about the other heterocyclic aromatic amines, heterocyclic phosphorous compounds or organometallic complexes as carcinogens or with cytostatic activity .
Examples of relevant publications are given below. It is recommended to the authors to cite these papers to give their introduction a wider base.
An unusual N-Alkylation Reaction during the Oxidative Addition of Hexafluoroacetone and Tetrachloro-o-benzoquinone to P-bis-(2-chloroethyl)-substituted l3P Compounds, I. Neda, C. Melnicky, A. Vollbrecht und R. Schmutzler, Synthesis, 1996, 473-474
Synthesis, structure, and reactivity of tetrakis(o,o-phosphorus)-bridged calix[4]resorcinols and their derivatives, Vollbrecht, A., Neda, I., Thonnessen, H., Jones, P.G., Harris, R.K., Crowe, L. A. Schmutzler, R., Chemische Berichte, 1997, 130, 1715-1720
Some references should be inserted.
In conclusion of my review
I recommend this manuscript for publication with minor revisions!
Author Response
- Reviewer #2:
1) The authors did not reflect any other field of other anti-cancer research, and other derivatives with the same anticancer activity, based on their antioxidant activity.
Response: Thank you for the suggestion. In the present study, we emphasized the protective role of Avn-C on anti-amyloid beta synaptotoxity in an Alzheimer’s Disease model mice. Thus, we excluded the description on the anti-cancer effect of Avn-C in the manuscript.
2) Still I believe that you should describe in the introduction more generally the synthesis and the use of phosphorous-organic compounds in general as alkylating or antioxidant agents. Therefore, you should cite more special topics. It is of interest for synthetic chemists which wide use have phosphorous-organic compounds. In the work of Schmutzler et al. could be found a huge variation of phosphorus organic compounds and their biological potential.
Response: We have extended the introduction part in the revised manuscript. Avn-C, as a natural compound from oats, has high antioxidant capacity and is not a phosphorus organic compound.
3) It is recommended to the authors to cite these papers to give their introduction a wider base.
An unusual N-Alkylation Reaction during the Oxidative Addition of Hexafluoroacetone and Tetrachloro-o-benzoquinone to P-bis-(2-chloroethyl)-substituted l3P Compounds, I. Neda, C. Melnicky, A. Vollbrecht und R. Schmutzler, Synthesis, 1996, 473-474
Synthesis, structure, and reactivity of tetrakis(o,o-phosphorus)-bridged calix[4]resorcinols and their derivatives, Vollbrecht, A., Neda, I., Thonnessen, H., Jones, P.G., Harris, R.K., Crowe, L. A. Schmutzler, R., Chemische Berichte, 1997, 130, 1715-1720
Some references should be inserted.
Response: Thank you for your comments. We have inserted several references in the introduction.
Reviewer 3 Report
In the manuscript entitled “Oat extract avenanthramide-C reverses hippocampal long-term 2 potentiation decline in Tg2576 mice” the authors characterise and discuss a compound isolated from oat, Avn-C, and its potential beneficial affect against AD. The authors did a lot of experiments, but in my opinion, they did not discuss in details the results. I recommend the publication of this manuscript only after major revisions.
- Introduction---I suggest to improve this paragraph
In the abstract the authors mention that “Memory deterioration in Alzheimer’s disease (AD) is thought to be underpinned by aberrant amyloid β (Aβ) accumulation, which contributes to synaptic plasticity impairment” (Line 18). I suggest to also add this statement in the first paragraph of the introduction, mentioning that recently in June 2021 Aducanumab (against Aβ) was approved by FDA for the treatment of AD.
-Canady, V.A. FDA Approves First Drug Therapy for Alzheimer’s in 18 Years. Mental Health Weekly 2021, 31, 3–4, doi:10.1002/mhw.32827.
Line 46: the authors start to describe the beneficial effect of oats. Before focusing the attention on the oat, I strongly recommend to insert few sentences in order to highlight the importance and the state of the art that the natural products have against AD progression for example:
-Ciccone L, Vandooren J, et al. Pharmaceuticals (Basel). 2021 Jan 24;14(2):86. doi:10.3390/ph14020086. PMID: 33498927; PMCID: PMC7911533.”
- Ferreira, João PS, et al. European Journal of Medicinal Chemistry (2021): 113492.
- Martins M, Silva R, M M Pinto M, Sousa E. Pharmaceuticals (Basel). 2020 Sep 11;13(9):242. doi:10.3390/ph13090242. PMID: 32933034; PMCID: PMC7558913
- Results
NMR---Line 145 In the description of NMR spectrum (1H, d, J=9.0 Hz, H-3), 7.52 (1H, d, J=3.2 Hz, H-6), 7.50 (1H, d, J=15.6 Hz, H-7'), 7.07 (1H, d, J=1.7 Hz, H-2'). 7.02 (1H, dd, J=9.0 and 3.2 Hz, H-4), 6.97 (1H, dd, J=8.2 and 1.7 Hz, H-6'), 6.79 (1H, d, J=8.2 Hz, H-5'), 6.46 (1H, d, J=15.6 Hz, H-8') you attributed each proton ex: H-6, H-7' …
- I suggest to add in the manuscript the chemical structure of avenanthramide C labelling the protons.
- Minor, in the supporting you could add the expansion of the NMR spectrum cutting the integral according to the NMR descripted in the text.
- Line 145 The authors wrote 1H-NMR (500 MHz) while in paragraph 3. Materials and Methods, it is written “Bruker Advance Digital 400 spectrometer… at 400 and 100 MHz. Please verify, and correct this unclear statement.
3.3.4 Total Phenol Quantification, DPPH, and ABTS.
The phenol quantification was performed in 96-well plate (Line 271), DPPH? in cuvette? (indicated volume 200+800 μL) Please clarify this point.
I recommend to change the paragraph Results in Results and Discussion, moreover I suggest to increase the discussion for each sub-session. Finally, paragraph 5. Conclusion need to be revised and extended according the discussion of the results.
Author Response
- Reviewer #3:
Introduction---I suggest to improve this paragraph
In the abstract the authors mention that “Memory deterioration in Alzheimer’s disease (AD) is thought to be underpinned by aberrant amyloid β (Aβ) accumulation, which contributes to synaptic plasticity impairment” (Line 18). I suggest to also add this statement in the first paragraph of the introduction, mentioning that recently in June 2021 Aducanumab (against Aβ) was approved by FDA for the treatment of AD.
-Canady, V.A. FDA Approves First Drug Therapy for Alzheimer’s in 18 Years. Mental Health Weekly 2021, 31, 3–4, doi:10.1002/mhw.32827.
Response: Thank you for your advice. As the reviewer’s suggestion, we have improved the introduction by including more information in the revised manuscript (page no. 1, line 38-43) (page no. 2, line 51-53).
Line 46: the authors start to describe the beneficial effect of oats. Before focusing the attention on the oat, I strongly recommend to insert few sentences in order to highlight the importance and the state of the art that the natural products have against AD progression for example:
-Ciccone L, Vandooren J, et al. Pharmaceuticals (Basel). 2021 Jan 24;14(2):86. doi:10.3390/ph14020086. PMID: 33498927; PMCID: PMC7911533.”
- Ferreira, João PS, et al. European Journal of Medicinal Chemistry (2021): 113492.
- Martins M, Silva R, M M Pinto M, Sousa E. Pharmaceuticals (Basel). 2020 Sep 11;13(9):242. doi:10.3390/ph13090242. PMID: 32933034; PMCID: PMC7558913
Response: We are grateful to reviewer for pointing out this problem. We have added it in revised manuscript (page no. 2, line 53-57).
- Results
I suggest to add in the manuscript the chemical structure of avenanthramide C labelling the protons.
Response: We have added a chemical structure of Avn-C as follows and it has been added in revised manuscript.
(page no. 6, added figure 4. line 157).
Minor, in the supporting you could add the expansion of the NMR spectrum cutting the integral according to the NMR descripted in the text.
Line 145 the authors wrote 1H-NMR (500 MHz) while in paragraph 3. Materials and Methods, it is written “Bruker Advance Digital 400 spectrometer… at 400 and 100 MHz. Please verify and correct this unclear statement.
Response: We apologize for this confusion. This section was revised (page no. 8, line 214-215).
3.3.4 Total Phenol Quantification, DPPH, and ABTS.
The phenol quantification was performed in 96-well plate (Line 271), DPPH? in cuvette? (indicated volume 200+800 μL) Please clarify this point.
Response: DPPH activity was performed in the cuvette. This section was revised in the manuscript (page no. 10, line 304).
I recommend to change the paragraph Results in Results and Discussion, moreover I suggest to increase the discussion for each sub-session. Finally, paragraph 5. Conclusion need to be revised and extended according the discussion of the results.
Response: We have changed the paragraph ‘Results’ in ‘Results and Discussion’ and added new information in the revised manuscript.
We have also extended the paragraph 4 ‘Conclusion” according to the reviewer’s comments (page no. 11-12, line 362-381).

Reviewer 4 Report
The manuscript “Oat extract avenanthramide-C reverses hippocampal long-term 2 potentiation decline in Tg2576 mice.” by Yu Young Lee et al. demonstrate the ability of Avn-C to increase synaptic plasticity in a mouse model of AD.
In my opinion, the study by itself seems appropriate and relevant, but some issues need to be corrected:
1- in the table 3 legend please describe the last column (Folin-Ciocalteau assay results).
2- In my opinion, the table 3 should be inserted at the end of the paragraph 2.2.
3- the Aβ-induced synaptic impairment via phospho-glycogen synthase kinase-3β (p-GSK3β-S9) and cleaved caspase 3 should be better described in the results and discussion section.
4- correct the formatting of the table legend.
Author Response
- Reviewer #4:
In my opinion, the study by itself seems appropriate and relevant, but some issues need to be corrected:
1- in the table 3 legend please describe the last column (Folin-Ciocalteau assay results).
Response: We are appreciating to reviewer for pointing out this problem. We have described this part in revised manuscript (page no. 4, line 122-124, and line 126).
2- In my opinion, the table 3 should be inserted at the end of the paragraph 2.2.
Response: We agree with the reviewer’s opinion. We have inserted the table 3 at the end of the paragraph 2.2 in the manuscript (page no. 3, line 121).
3- the Aβ-induced synaptic impairment via phospho-glycogen synthase kinase-3β (p-GSK3β-S9) and cleaved caspase 3 should be better described in the results and discussion section.
Response: As the reviewer’s suggestion, we have included this description in the manuscript (page no. 8, line 194-201)
4- correct the formatting of the table legend.
Response: Format of the table legend has been changed in the manuscript.
Round 2
Reviewer 3 Report
Dear authors,
I am glad to verify that you answered point by point at all the remarks.
I recommend the publication of your revised manuscript.
Sincerely yours